# Does Direct Monetary Policy Affect the Supply of Bank Credit to Small and Medium-Sized Enterprises? An Analysis Based on Chinese Data

**Ruishi Jiang**  **and Jia Ruan \***

School of Economics and Management, Beijing Jiaotong University, Beijing 100044, China; 22110169@bjtu.edu.cn
* Correspondence: jruan@bjtu.edu.cn

**Abstract:** In order to develop the real economy and solve the problems of enterprise financing and lending, banks should increase their support for SMEs (small and medium-sized enterprises). The People's Bank of China introduced two direct monetary policy tools in June 2020, which are important for alleviating the financing problems of SMEs, improving the construction of financial support for the real economy and promoting the recovery of economic development. This paper manually collects annual data on the loan balances of SMEs from listed commercial banks in China from 2011 to 2021, and it empirically tests the implementation effects of the direct monetary policy tools using the double-difference model and the moderating effect model. The results of the study indicate that the implementation of the direct monetary policy tools can increase the credit supply of commercial banks to SMEs. The moderating effect of digital inclusive finance on the impact of direct monetary policy on the credit supply of SMEs is not statistically significant for the time being. Therefore, it is necessary to improve the transmission mechanism of structural monetary policy, establish a sustainable development mechanism for digital inclusive finance, and guide commercial banks to improve their capital strength and profitability so that structural monetary policy tools can be most effective.

**Keywords:** structural monetary policy; supply of bank credit to SMEs; sustainability of SMEs; digital inclusive finance; double-difference model

## 1. Introduction

SMEs are an important vehicle for China to stabilize economic growth, promote innovative practices, increase employment and improve people's livelihood. As of 2022, the number of private enterprises in China has grown to 47,011,000, and the share of private enterprises in the total number of enterprises has increased to 93.3%, contributing to more than 50% of China's tax revenue and more than 60% of its GDP. Therefore, China attaches great importance to the development of SMEs and requires banks and other financial institutions to increase their support for SMEs to promote the recovery of the real economy. Data from the CBRC (China Banking Regulatory Commission) show that as of the end of 2022, the balance of loans used by banking financial institutions for SMEs was CNY 59.7 trillion, including CNY 23.6 trillion of loans for inclusive SMEs with a total single-account credit of CNY 10 million or less, with a year-on-year growth rate of 23.6%. The increase in bank loans for inclusive SMEs improved the availability of bank credit for SMEs, reduced operating costs for SMEs, and allowed SMEs to continue their operations. The supply of bank credit to SMEs cannot be separated from digital inclusive finance [1–3]. Banks and other financial institutions should invest more in digital transformation, accelerate digital transformation, and continuously improve the level of fintech application. Digital inclusive finance has become an important development direction for China's financial economy in the new era, and it is an important contributor

to the commercial sustainability of banks and the healthy and stable development of the economy [4,5].

Since 2020, China has used structural monetary policy tools to vigorously promote the development of inclusive finance in the banking sector by creating monetary policy measures such as the loan extension support tool for inclusive SMEs, the credit loan support program for inclusive SMEs and inclusive finance refinancing. The purposes of these policy tools are to encourage banks to strengthen the application of financial technology tools under the premise of commercial sustainability, strengthen loan risk prevention through digital inclusive finance, increase the allocation of loans for inclusive SMEs, promote commercial banks to accelerate the formation of a mechanism that dares to lend, can lend and will lend, alleviate the financing problems of SMEs, accelerate the digital transformation of banks, enhance the ability of banks and other financial institutions to serve SMEs, and promote finance in the new stage of development, serving the real economy better. The "direct access to the real economy" is targeted and efficient, which means that banks should shorten the transmission path of monetary policy and improve the efficiency of monetary policy transmission. The direct monetary policy tool is intended to supplement traditional bank-based monetary policy transmission to address the financing difficulties of SMEs, to leverage financial technology, to bring into play the effectiveness of digital inclusive finance, to improve the efficiency of monetary policy transmission, and to promote the resumption of work and digital transformation of SMEs.

SMEs are an important foundation for economic and social development and an important pillar of the national economy. Affected by the COVID-19 pandemic, the operation of SMEs is stagnant, the financing channel is single, the financing is difficult, and it is difficult to maintain operation. The People's Bank of China has introduced two direct monetary policy tools in response to the financing problems of SMEs. Whether or not SMEs are easier to finance due to the newly introduced monetary policy, the effect of the implementation of monetary policy is worth exploring. If the direct monetary policy has a good effect, it can imitate the operating mechanism of this monetary policy tool and introduce a new monetary policy tool to other areas of the real economy. Applying incentive compatibility theory, information asymmetry theory and credit rationing theory, this paper manually collects annual data on the loan balances of SMEs from listed commercial banks in China from 2011 to 2021, and it empirically tests the implementation effects of the direct monetary policy tools using the double-difference model and the moderating effect model.

Compared with the previous literature, the possible innovations and contributions of this paper are mainly in two aspects. First, few studies have been conducted on the newly introduced structural monetary policy tools such as direct monetary policy. This paper has a relatively new research sample, and it is of good practical significance to manually collect the SME loan balances of Chinese listed commercial banks from 2011 to 2021 and empirically analyze the implementation effects of direct monetary policy tools. The operating mechanism of these two monetary policy tools provides a reference for the formulation of new monetary policy instruments. Second, few existing studies have integrated the two perspectives of digital inclusive finance and direct monetary policy. This paper uses a double-difference model to assess the effectiveness of direct monetary policy tools and uses digital inclusive finance as a moderating effect variable to study the policy implementation effect.

## 2. Literature Review

### 2.1. Impact of Structural Monetary Policy Tools on the Supply of Credit to SMEs

The traditional monetary policy based on aggregate regulation cannot effectively solve the financing problems in key areas and weak links of the national economy. First, most of the funds released by the aggregate monetary policy flow to state-owned enterprises and the real estate industry, making it difficult for funds to flow to SMEs. Second, it is difficult for the aggregate monetary policy to reduce the financing costs of SMEs. Traditional monetary policy cannot effectively guide banks and other financial institutions in the

investment of credit funds, and it does not support the supply of credit for SMEs. The structural monetary policy can effectively make up for the defects of traditional aggregate monetary policy, reduce the cost of enterprise financing, guide the flow of credit funds to downstream SMEs, promote the development of target enterprises, adjust the operating costs and credit structure of financial institutions, play a greater role in financing SMEs, and better achieve the goal of maximizing social welfare. To make up for the shortcomings of traditional monetary policy, the People's Bank of China has innovated and implemented a series of structural monetary policy tools since 2013, such as targeted downgrades, small refinancing, rediscount guidance, standing lending facilities (SLF), targeted medium-term lending facilities (TMLF), and collateralized supplementary loans (PSL), to alleviate the financing difficulties of SMEs.

Among the many structural monetary policy tools in China, targeted downgrading is the most representative and the most studied one. The targeted downgrade policy is transmitted through the path of "central bank–commercial banks–targeted support enterprises" and mainly works on the real economy through the credit channel. Many scholars have conducted studies of the effects of the targeted downgrade policy. Targeted downgrading improves the availability of credit and reduces the financing cost of targeted support areas such as SMEs, which can better improve the financing situation of SMEs. Commercial banks adjust their credit supply structure to meet the proportion of credit allocation to SMEs required by the People's Bank of China's targeted downgrade in order to obtain the preferential policy of targeted downgrade, and the supply of credit funds to large and medium-sized enterprises decreases and the supply of credit funds to SMEs increases, which effectively enhances financial institutions' preference for credit allocation to SMEs and motivates commercial banks to allocate more credit resources to SMEs [6], and the increase in the number of SME loans by commercial banks also has a spillover effect on the credit access of non-SMEs. The increased availability of loans to SMEs can significantly reduce their demand for commercial credit, resulting in a significant "financing constraint relief effect" and "investment stimulation effect", which can promote the expansion of investment in SMEs by reducing their financing costs. Through the reciprocal symbiotic effect between enterprises, the development of SMEs can drive the growth of the output of large enterprises, and the implementation of preferential corporate income tax policies can produce a "1 + 1 > 2" superimposed universal benefit effect.

In the USA, the Federal Reserve's Operation Twist (OT) and Term Loan Auction Facility (TAF) also influence the flow of funds through regulating interest rates. The Fed's Operation Twist (OT) uses the open market to sell large amounts of treasuries and hold medium-term and long-term treasuries to lower long-term interest rates, direct liquidity toward long-term assets, increase long-term liquidity for financial institutions, improve the asset mix of commercial banks, and achieve a reduction in long-term financing costs for the real economy [7–9]. The US Term Loan Auction Facility (TAF) has an impact on Libor–OIS spreads, and the TAF can significantly alleviate liquidity pressures in the interbank market, easing the financing difficulties and expensive financing for the real economy [10].

As a typical direct monetary policy tool for the real economy, refinancing supports the resumption of work and production of SMEs that are less resilient to the COVID-19 pandemic. From the empirical study of the effect of the refinancing policy, under the reporting model of "lending before borrowing", banks and other financial institutions first issue credit funds to SMEs that meet the policy requirements, and they then apply for refinancing funds from the People's Bank of China in equal amounts, which directly increases the supply of credit to SMEs and reduces the interest rate of commercial banks for SMEs. This directly increases the supply of credit to SMEs, reduces the interest rate of loans issued by commercial banks to SMEs, and reduces the financing cost of SMEs. The small-scale refinancing promotes the increase in banks' general micro and small loans, although there is an optimal size range of structural monetary policy instruments for small-scale refinancing. The structural monetary policy exemplified by targeted downgrades and refinancing has led to banks' lending preference for SMEs, forming a positive incentive

mechanism, increased credit allocation to SMEs, and targeted and precise guidance of capital flow to SMEs to alleviate their financing dilemmas, the transmission effect of which has been enhanced with the maturity of structural monetary policy.

In Britain, the ECBC and the Bank of Scotland also implemented a series of refinancing operations to boost the supply of corporate credit. The targeted long-term refinancing operations (TLTRO) of the ECBC have a significant targeting effect, with the TLTRO increasing the amount of credit to the real economy [11], improving the real economy and effectively contributing to growth of targeted lending, with limited spillover effects on non-targeted lending [12,13]. The increase in the number of targeted loans following the Bank of England's Financing for Lending Scheme (FLS) drove commercial bank credit growth, effectively supporting the real economy [14]. The European People's Bank of China's long-term refinancing operations (VLTROs) had a positive economic impact on bank and real economy lending by extending bank debt maturities during the crisis period, with positive effects on lending to Spain, Italy and Portugal, among other countries [15–17], making a positive contribution to targeted corporate lending, which can effectively boost credit supply and reduce the risk of credit default.

Many scholars are skeptical about the effectiveness of the implementation of direct monetary policy. The Fed's Operation Twist (OT) has a significant effect on the Treasury bill market but a diminished effect on the improvement of private sector credit. It cannot fundamentally reduce long-term interest rates and stimulate an increase in credit liquidity, and other financing channels are needed to reduce the cost of corporate finance [18]. The Fed's Term Loan Auction Facility (TAF) has also not been effective in reducing spreads [19]. The ECBC's Targeted Long-Term Refinancing Operation (TLTRO), which can stimulate fixed asset investment in large firms by reducing credit constraints, is difficult to be effective in the face of conditions such as poor corporate asset quality and insufficient collateral, and it does not have a significant effect on alleviating credit constraints for micro and small firms [20,21]. The European Central Bank's long-term refinancing operations (VLTRO) have increased the credit risk of commercial banks, and the injected liquidity has led to an increase in the holdings of high-yield government bonds [15,22], which reduces the operational performance of banks and affects the supply of credit to the real economy [17,23]. The Bank of England's Finance for Lending Scheme (FLS) had a non-significant effect on increasing the number of loans to SMEs, and the decline in the cost of financing in the credit market affected the effectiveness of the Bank's policy to implement FLS [24].

In the current financial environment, there are still some problems in terms of the effectiveness and accuracy of direct monetary policy. The funds released under the influence of the policy cannot be guaranteed to flow to the target areas. The effect is greatly different from expectations. It does not promote the release of funds to SMEs; instead, it leads to the inflow of liquidity to non-target enterprises and cannot serve to alleviate the financing difficulties of SMEs. The operating mode and experience of major structural monetary policy tools such as financing-to-loan plans, targeted long-term refinancing operations, and fixed-term loan auctions in Europe and the United States are worth learning from. China should enrich the types of monetary policy tools and create new monetary policy tools. The economy stagnates in the short term due to the outbreak of the COVID-19 pandemic in 2020. The severe external macro environment has exacerbated the uncertainty of economic development. In China, the poor ability of SMEs to withstand risks and the deterioration of the economic liquidity environment have affected the operation of SMEs with capital shortage and financing difficulties, for which the People's Bank of China created two direct monetary policy tools to help SMEs alleviate their financing difficulties.

### 2.2. The Impact of Digital Inclusive Finance on the Supply of Credit to SMEs

Digital inclusive finance is a new model of inclusive finance developed on the basis of digital technology. Through the organic integration of digital technology and inclusive finance, it can expand the reach of finance, break through the limitations of time and space, improve the quality of financial services, expand the scope of financial services,

improve the coverage and accessibility of financial services, promote the efficiency of the real economy of financial services, and play an important role in improving the efficiency of financing for SMEs, broadening financing channels and reducing financing costs. It plays an important role for SMEs in improving the efficiency of financing, broadening financing channels and reducing financing costs. Digital inclusive finance continuously increases the breadth and depth of financial services with the help of digital technologies such as blockchain, big data and cloud computing; efficiently and accurately collects and mines multidimensional soft information such as account flow, dynamic revenue and historical transactions of SMEs [25]; improves information selection and risk identification [26]; and reduces information collection and processing costs, capital transaction costs and credit risk assessment costs, effectively resolving the information asymmetry between financial institutions and SMEs, and improving the availability of financing for SMEs [27,28]. The development of digital inclusive finance has improved SMEs' access to credit resources from banks and other formal financial institutions, increased the scale of financing, and reduced the cost of financing, which to a certain extent alleviates the pressure on SMEs in terms of financing and expensive financing.

Little domestic literature has studied the practical impact of digital inclusive finance on the credit supply problem of SMEs. The existing domestic literature focuses on the role of digital inclusive finance in alleviating financing constraints [29–33], and the financing constraints play a mediating role in achieving digital inclusive finance for business innovation. By incorporating digital technology into all aspects of SME financing, digital inclusive finance effectively compensates for the shortcomings of traditional financial institutions, realizes the tripartite linkage between SMEs, banks and other financial institutions, and the government, and breaks the disadvantageous barriers facing SMEs in traditional financing. Digital inclusive finance can also lower the service threshold by providing diversified financial services, enhance the accessibility of services, reduce financing costs, ease the financing constraints of SMEs, promote technological innovation by opening up new financing channels, promote green innovation in enterprises, improve the level of innovation, reduce the leverage of SMEs, promote the improvement of the total factor productivity of SMEs, and enhance the value of enterprises in the long run. In regions with poorly developed banking and capital market sectors, digital inclusive finance can effectively alleviate the financing constraints of small-scale enterprises and high-tech enterprises. The alleviation effect of digital inclusive finance on enterprise financing constraints will rise with the increase in the CSR level.

### 2.3. Digital Inclusive Finance Affects the Transmission of Direct Monetary Policy

The transmission mechanism of structural monetary policy mainly works in three ways. First, the People's Bank of China lends to banks that meet the policy requirements, targeting liquidity funds to specific areas, increasing the supply of low interest rate policy funds, lowering the cost of bank loans, and thus reducing the cost of financing for the real economy. Second, the People's Bank of China reduces the legal reserve ratio through differentiated deposit reserves, increasing the amount of funds available for banks to lend, thereby increasing credit support to the real economy. Third, the People's Bank of China, with the help of direct policy funding support signals, introduces incentive-compatible mechanisms to mobilize banks and guide them to adjust their credit structure and change their credit allocation decisions, thus promoting the expansion of credit allocation to the real economy.

Developing digital inclusive finance is an inevitable choice for banks' sustainable development. Commercial banks have responded to the call of monetary policy and have vigorously developed inclusive finance to effectively alleviate the financing difficulties of SMEs and increase the effective supply of finance to real enterprises. In June 2020, the People's Bank of China created the Inclusive SMEs Loan Deferred Principal and Interest Repayment Support Tool and Inclusive SMEs Credit Loan Support Program to provide preferential funding to incentivize banks and other financial institutions to provide a credit

supply for SMEs. In order to better utilize the implementation effect of direct monetary policy, commercial banks undergo digital transformation, establish departments dedicated to serving SMEs, obtain more information on SMEs with the help of digital inclusive finance, develop financial products and services applicable to SMEs, and guarantee a credit supply to SMEs (Buchak et al., 2018) [34]. Commercial banks set up inclusive finance divisions and implement small and inclusive finance business assessment mechanisms, which can better respond to various national support policies for SMEs, increase credit rationing channels, better allocate various credit resources, reduce banks' risk-taking, greatly improve the availability of credit for SMEs, and also provide an information platform for SMEs to obtain national support policies.

In summary, the purpose of this literature review is to summarize the implementation effect of monetary policy tools of the real economy, such as direct enterprises from different countries in different periods. The literature study found that the implementation effect of monetary policy is not necessarily effective. Two direct monetary policies have just been introduced by the central bank after the COVID-19 pandemic. Unlike monetary policy tools such as refinancing and rediscounting, they are a supplement to structural monetary policy tools. There are more studies on the policy effects of structural monetary policy tools such as targeted downgrades and refinancing in the domestic and international literature, and fewer studies were conducted on the effects of the implementation of direct monetary policy tools. What is the rationale behind the direct monetary policy tool and its operating mechanism? Can the implementation of this policy tool effectively alleviate the problem of difficult and expensive financing for SMEs and increase the supply of credit from banks to SMEs? Does digital inclusive finance play a moderating role in the process of direct monetary policy influencing banks' credit supply to SMEs? This paper will focus on assessing the effect of innovative direct-to-real-economy monetary policy tools to provide suggestions for further improving structural monetary policy tools, alleviating credit pressure on SMEs, and better developing the real economy.

## 3. Mechanism of Direct Monetary Policy Instruments

### 3.1. Inclusive SMEs Loan Deferred Principal and Interest Repayment Support Tool

In order to encourage specific local banks to extend loans to SMEs and reduce the pressure on capital repayment funds for SMEs, the People's Bank of China has launched CNY 40 billion loan extension support tool for all the SMEs. The six types of specific local banks which implement this monetary policy include urban commercial banks, rural commercial banks, rural cooperative banks, rural credit cooperatives, village banks and private banks (including Internet banks), which meet the policy requirements. Through the Special Purpose Vehicle (SPV), the People's Bank of China (PBC) enters into an interest rate swap agreement with specific local banks to incentivize them to increase the supply of credit for deferred debt service for small and medium-sized enterprises, and specific local banks receive 1% of the principal amount of interest-free funds from the PBC to compensate for the loss caused by the deferred debt service, as shown in Figure 1. This instrument is expected to support specific local banks to defer debt service loan principal of CNY 4 trillion. If there are multiple extensions of principal repayment for the same loan within a certain period of time, the incentive applies only to the first extension. The credit risk of the corresponding loan is still borne by the local corporate bank to avoid moral hazards.

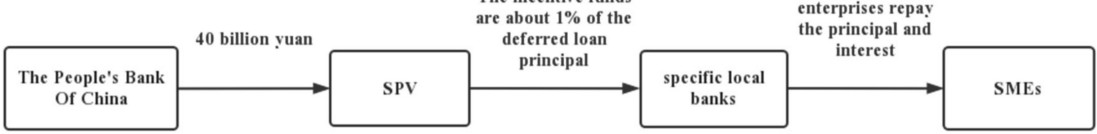

**Figure 1.** Mechanism of the Inclusive SMEs Loan Deferred Principal and Interest Repayment Support Tool.

### *3.2. Inclusive SMEs Credit Loan Support Program*

Compared with large state-owned enterprises, SMEs are riskier to operate, and banks will prefer small and micro loans with collateral or guarantee for risk prevention reasons. In order to alleviate the problem of difficult and expensive financing caused by the lack of collateral for SMEs and encourage banks to increase credit funds for SMEs, the People's Bank of China has created the CNY 400 billion Inclusive SMEs Credit Loan Support Program. The loan is the newly issued credit loan with a term of not less than 6 months from 1 March to 31 December 2020 to eligible specific local banks (the same as the scope of the implementation of the Inclusive SMEs Loan Deferred Principal and Interest Repayment Support Tool) for all kinds of SMEs. Through the SPV, the PBC and the specific local banks sign a contract for the credit loan support program, and after the PBC purchases the credit loans through monetary policy instruments, it entrusts them to the lending banks for management. The People's Bank of China provides zero-interest rate concessional funds for a period of one year at 40% of the principal amount of credit loans actually issued by specific local banks, as shown in Figure 2. This tool is expected to issue CNY 1 trillion of new credit loans for inclusive SMEs.

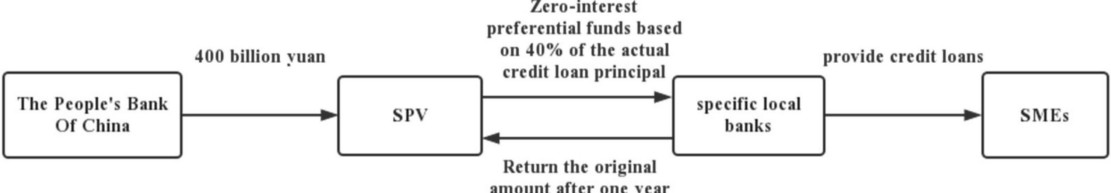

**Figure 2.** Mechanism of the Inclusive SMEs Credit Loan Support Program.

The two direct monetary policies have a time limit. On 6 April 2021, the People's Bank of China issued a document to extend the implementation period of the two direct monetary policy tools to the end of 2021. The People's Bank of China has done a good job in the policy succession and conversion of the two direct tools. Firstly, for the implementation of the inclusive small and micro loan support tool, financial institutions and enterprises independently negotiated loan repayment and interest payment in accordance with the market-based principle. From 2022 to the end of June 2023, the People's Bank of China has provided funding for the loans issued by the local legal entity banks to inclusive SMEs and individual business households, according to 1% of the incremental balance, to encourage the increase in inclusive small and micro loans. Secondly, inclusive small and micro-credit loans will be included in the management of the support program for agriculture and small and medium-sized refinancing, and the original CNY 400 billion of the refinancing quota used to support inclusive small and micro-credit loans can be used on a rolling basis, and the refinancing quota can be further increased if necessary.

Indirect financing, mainly from the banking system, is the main source of financing for SMEs in China. After the People's Bank of China issues monetary policy, commercial banks act as the transmission hub, and the direct monetary policy tools do not go beyond banks and other financial institutions to reach SMEs directly. The "direct access" means that the transmission path of monetary policy is shortened, the transmission channel of monetary policy is smooth, the specificity of monetary policy tools to support various types of enterprises is enhanced, and the role of monetary policy to serve the real economy is better played. The direct access monetary policy tools are set up through the SPV to ensure the efficiency of monetary policy effects and stimulate the vitality of market players.

### 4. Research Hypothesis

#### *4.1. Incentive Compatibility Theory*

Due to the small size of SMEs, lack of assets, opaque financial information, imperfect management system and high business risks, commercial banks have difficulties in obtaining effective information to assess their risks and are reluctant to provide financial

support to SMEs due to their own lending risks and other issues. In order to enhance banks' preference for credit investment in SMEs, and to accept to a greater extent the non-performing loans generated by lending to SMEs, the PBC needs to introduce some policies to provide incentives to banks. The incentives are divided into positive and negative incentives. On the one hand, negative incentives are set for banks by setting assessment indicators, requiring commercial banks to provide a certain percentage of their funds for loans to SMEs every month, and they will be punished if they fail to meet the requirements of the indicators. This kind of negative incentive policy is detrimental to the interests of commercial banks, and in the long run, commercial banks may respond to such policies in a negative way, leading to more serious adverse consequences. On the other hand, a positive incentive policy tool can be used by commercial banks to provide targeted preferential funds and to guide them to flow funds to specific areas of SMEs, which not only realizes the subjective regulation intention of the PBC but also satisfies the capital needs of commercial banks themselves, so that both the PBC and commercial banks can maximize their utility.

The two direct monetary policy tools created by the People's Bank of China contain incentive-compatible theoretical mechanisms. The Inclusive SMEs Loan Deferred Principal and Interest Repayment Support Tool supports financial institutions to defer capital and interest payments on loans to SMEs that temporarily encounter difficulties, and it provides open-ended interest-free funding for 1% of the principal amount of deferred loans to specific local banks in response to the losses incurred from deferred capital payments. The Inclusive SMEs Credit Loan Support Program provides one-year zero-interest funds at 40% of the actual principal amount of credit loans granted by specific local banks to increase the supply of credit funds to SMEs by specific local banks. With the incentive compatibility mechanism, commercial banks will increase credit allocation to SMEs in order to continue to receive preferential funding support, while also allowing the People's Bank of China to reach its subjective regulatory intent to direct the flow of funds to specific areas such as SMEs, bringing into play the efficient and direct nature of direct access monetary policy tools.

**Hypothesis 1.** *The direct monetary policy instrument increases bank credit allocation to SMEs by easing the pressure of non-performing loans and enhancing the liquidity of targeted loans to SMEs through the incentives provided by the People's Bank of China to specific local banks.*

*4.2. Information Asymmetry Theory*

Digital inclusive finance can effectively alleviate the information asymmetry between banks and enterprises, avoid adverse selection and moral hazards, and improve the financing availability and efficiency of SMEs. When traditional financial institutions, as represented by banks, review the loan demand projects of SMEs, they should strictly review the process and assess the risks of SMEs in an all-round and multi-level manner in order to avoid default events.

With the help of digital technologies such as artificial intelligence, blockchain, big data and cloud computing, digital inclusive finance can establish information monitoring and processing systems and risk control systems, which can efficiently and accurately collect and process multi-dimensional soft information on the financial status, actual operation and ability to resist risks of SMEs, and they can effectively reduce risk assessment costs and transaction costs. This can also simplify the process of enterprise qualification review and loan approval, reduce credit review time, lower the cost of asset credit assessment, offline review and risk management, and provide diversified credit support and financial services for many SMEs that lack traditional financing qualifications [35,36].

**Hypothesis 2a.** *The development of digital inclusive finance is conducive to improving commercial banks' information collection and risk assessment of SMEs, alleviating the information asymmetry and increasing the credit supply to SMEs.*

*4.3. Credit Rationing Theory*

Compared with large and medium-sized enterprises, SMEs operate on a small scale, have opaque financial statements and unsound financial structure, and their credit records lack guarantees or collaterals, which makes it difficult for banks and other financial institutions to understand the real financing purpose, property status and management level of SMEs. In order to reduce their own risk and prevent the occurrence of non-performing loans, banks are "credit discriminatory" when lending to enterprises [37], preferring to provide credit funds to large and medium-sized enterprises with good credit standing, while lending to SMEs with strict qualifications, leading to an increase in the cost of lending for banks and other financial institutions. This leads to an increase in the cost of loans for banks and other financial institutions, and banks pass on the cost to SMEs by raising credit rates and increasing the cost of loans for SMEs, which become the target of credit rationing by banks [38].

Relying on big data, digital inclusive finance uses text mining technology to collect multi-dimensional data such as customer credit, conduct information matching and deep-level mining analysis, conduct credit portrait and credit evaluation of target enterprises, and integrate the third-party data of individuals, enterprises and industries. If the third-party credit system can be established and improved, the cost of financial services will be reduced, the scope of financial services expanded, and financial services will be more convenient and efficient.

**Hypothesis 2b.** *The development of digital inclusive finance is universal, and large and medium-sized enterprises have easier access to bank credit funds and financial services compared to SMEs. The existence of the problem with credit rationing makes the moderating effect of digital inclusive finance on the bank credit supply to SMEs insignificant.*

## 5. Study Design

*5.1. Data Sources*

The credit supply of commercial banks to SMEs after the implementation of direct monetary policy is examined. This paper manually collects data on banks' SME loan balances from the annual reports on the official websites of 174 Chinese listed commercial banks from 2011 to 2021 as the research object. Other data are obtained from the Wind database and CSMAR database.

*5.2. Model Setting*

The People's Bank of China created two direct monetary policy tools on 1 June 2020, which are the Inclusive Small and Micro Enterprise Loan Deferred Principal and Interest Repayment Support Tool and the Inclusive Small and Micro Enterprise Credit Loan Support Program. To test the implementation effect of these monetary policy tools, this paper establishes a double-difference model (1), with specific local banks that meet the policy requirements and enjoy the policy preferences as the experimental group and commercial banks that do not meet the policy requirements and do not enjoy the policy preferences as the control group.

$$SMEL_{i,t} = \beta_0 + \beta_1 DID + \beta_2 policy + \beta_3 treat + \gamma control + \delta + \mu + \varepsilon \qquad (1)$$

To test the moderating effect of digital inclusive finance on direct monetary policy affecting credit supply to SMEs, model (2) is constructed.

$$SMEL_{i,t} = \beta_0 + \beta_1 DID + \beta_2 policy + \beta_3 treat + \beta_4 df + \beta_5 DID * df + \gamma control + \delta + \mu + \varepsilon \quad (2)$$

*SMEL* is the explanatory variable for the share of inclusive SME loans of commercial banks, *DID* is the double-difference variable to be focused on in this paper to measure the effect of the implementation of direct monetary policy tools, *policy* indicates that it takes the value of 1 before 2020 and the value of 0 after 2020, *treat* indicates that the

eligible specific local banks, i.e., the experimental group, take the value of 1, while the commercial banks that do not enjoy the policy commercial banks, i.e., the control group, take the value of 0, *df* denotes the digital inclusive finance variable, *DID* ∗ *df* denotes the moderating effect variable of digital inclusive finance affecting the effect of direct monetary policy, *control* denotes all the control variables of commercial banks, $\delta$ denotes the fixed effect of commercial banks, $\mu$ denotes the time fixed effect, and $\varepsilon$ denotes the random disturbance term.

### 5.3. Variable Selection

#### 5.3.1. Explanatory Variables

The supply of credit to SMEs (*SMEL*) is the explanatory variable in this paper, and it is expressed as a percentage of commercial banks' SME loan balances in their total loans after standardization of the Z-score.

#### 5.3.2. Core Explanatory Variables

The explanatory variable (*DID*) consists of the cross product of whether or not the direct monetary policy is implemented (*policy*) and whether or not the bank is a locally incorporated bank enjoying the direct monetary policy (*treat*). If the sample time is June 2020 and later, policy = 1; otherwise, policy = 0. If the commercial bank is a local legal person bank that enjoys direct monetary policy, treat = 1; otherwise, treat = 0.

The moderating variable is the Digital Inclusive Finance Index of Peking University, and this paper matches the addresses of commercial banks with the Digital Inclusive Finance Index of each prefecture-level city and selects the total index of the Digital Inclusive Finance Index (*index_aggregate*) and three sub-indicators, digital financial coverage breadth (*coverage_breadth*), digital financial usage depth (*usage_depth*) and digitization level (*digitization_level*), and the *DID* intersection multiplier.

#### 5.3.3. Other Control Variables

In order to reduce the problem of multicollinearity and test the correlation between variables, the following six variables are selected as control variables of the model.

Here, *ttloan* is expressed as the logarithm of the total loans of commercial banks, *loan_asset_ratio* is expressed as the proportion of the total loans to the total assets of commercial banks, *depositgrowth* is expressed as the proportion of the difference between the current and previous deposits of commercial banks to previous deposits, *asset-liability_ratio* is expressed as the proportion of total liabilities of commercial banks to previous deposits, *asset-liability_ratio* is expressed as the proportion of total liabilities of commercial banks to previous deposits, *depositgrowth* is expressed as the difference between current and previous deposits of commercial banks as a proportion of previous deposits, *asset_liability_ratio* is expressed as the proportion of total liabilities of commercial banks to total assets, *rota* is expressed as the proportion of net profit of commercial banks to total assets, and *nplra* is expressed as the amount of non-performing loans of commercial banks as a proportion of the total loans. The definitions and descriptions of the variables are shown in Table 1.

**Table 1.** Definitions and descriptions of variables.

| Variable Name | Variable Symbol | Variable Description |
|---|---|---|
| Credit supply for SMEs | *SMEL* | Balance of bank loans to SME/total loans |
| Digital Inclusive Finance Index | *index_aggregate* | |
| Breadth of digital financial coverage | *coverage_breadth* | Peking University in the prefecture where the |
| Depth of digital finance usage | *usage_depth* | commercial bank is located |
| Digitalization of financial inclusion | *digitization_level* | Digital Inclusive Finance Index |
| Digital finance usage depth of Credit Index | *credit* | |
| Total loans | *ttloan* | Total commercial bank loans |

**Table 1.** *Cont.*

| Variable Name | Variable Symbol | Variable Description |
|---|---|---|
| Loan-to-asset ratio | *loan_asset_ratio* | Total loans/total assets |
| Deposit growth rate | *depositgrowth* | (Current deposit − previous deposit)/previous deposit |
| Asset liability ratio | *asset_liability_ratio* | Total liabilities/total assets |
| Return on total assets | *rota* | Net profit/total assets |
| Non-performing loan ratio | *nplra* | Amount of non-performing loans/total loans |
| The effect of the implementation of direct monetary policy tools | *DID* | Policy ∗ treat |
| Whether or not the direct monetary policy is implemented | *policy* | The value of 1 before 2020 and the value of 0 after 2020 |
| Whether or not the bank is a locally incorporated bank enjoying the direct monetary policy | *treat* | The experimental group takes the value of 1 and the control group takes the value of 0 |

## 6. Empirical Analysis

### 6.1. Descriptive Statistics

The results of the descriptive statistics concerning the main variables in this paper are shown in Table 2. Among them is the supply of credit to SMEs (*SMEL*), expressed as the proportion of the balance of loans to SMEs of commercial banks in their total loans after the standardization of the Z-score, so the mean value is 0 and the standard deviation is 1. There are large differences in the supply of credit to SMEs of banks, Digital Inclusive Finance Index and other financial data of commercial banks, etc. The range of values of all the variables is basically reasonable, with sample distribution.

**Table 2.** Descriptive statistics of all the variables.

| Variables | Sample Size | Average Value | Standard Deviation | Minimum Value | Maximum Value |
|---|---|---|---|---|---|
| *SMEL* | 910 | $7.89 \times 10^{-10}$ | 1 | −0.3302695 | 29.18046 |
| *index_aggregate* | 910 | 240.2284 | 65.0157 | 45.66 | 359.68 |
| *coverage_breadth* | 910 | 239.0887 | 64.49856 | 41.49 | 371.79 |
| *usage_depth* | 910 | 235.3482 | 67.02201 | 48.05 | 354.3 |
| *digitization_level* | 910 | 252.8711 | 75.16899 | 13.94 | 340.01 |
| *credit* | 910 | 158.6618 | 37.25061 | 8.58 | 215.15 |
| *ttloan* | 910 | 25.5396 | 1.844891 | 21.06985 | 30.55549 |
| *loan_asset_ratio* | 908 | 0.5653524 | 0.1171322 | 0.01 | 1.1 |
| *depositgrowth* | 903 | 40.4485 | 11.61157 | 1 | 102 |
| *asset_liability_ratio* | 908 | 0.9255947 | 0.0200764 | 0.69 | 1.01 |
| *rota* | 908 | 5.22467 | 0.7956382 | 1 | 13 |
| *nplra* | 910 | 1.624418 | 1.042251 | 0 | 13.89 |

### 6.2. Parallel Trend Test

The double-difference model requires that the common trend assumption is satisfied, i.e., the outcome variable indicates the same trend in the experimental and control groups before policy implementation. In order to test the effect of the implementation of the two direct monetary policy instruments on the credit supply of SMEs better, the parallel trend of the credit supply to SMEs in the experimental and control groups before and after enjoying the direct monetary policy is tested. The parallel trend test is shown in Figure 3. There is no significant difference between the credit supply of SMEs in the experimental and control groups before and after the policy point in time (the significance level is 10%), which can pass the parallel trend test.

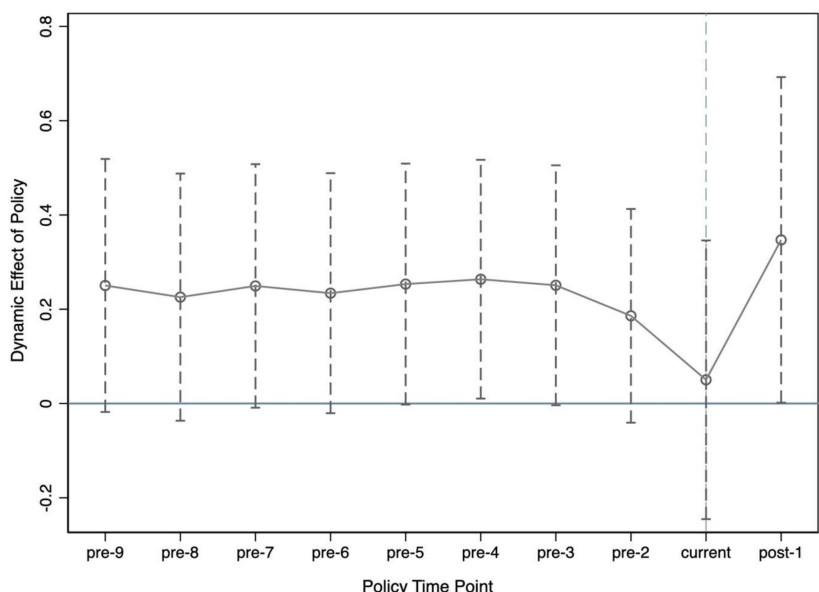

**Figure 3.** Parallel trend test.

### 6.3. A Test of the Effectiveness of the Implementation of Direct Monetary Policy

Based on the double-difference model identification framework constructed earlier, the implementation effects of the two direct monetary policy instruments are examined with the SMEs credit supply (*SMEL*) as the explanatory variable, where *DID* indicates the implementation effects of direct monetary policy instruments affecting SMEs credit supply. The specific regression results are shown in Table 3, where column (1) is the baseline regression without controlling for individual and time effects, column (2) is the baseline regression controlling for individual and time effects, and column (3) is the Tobit regression.

**Table 3.** Impact of direct monetary policy on the supply of credit to SMEs by banks.

|  | (1) | (2) | (3) |
|---|---|---|---|
|  | *SMEL* | *SMEL* | *SMEL* |
| *DID* | 0.2403 *** | 0.3519 ** | 0.1472 * |
|  | (0.0723) | (0.1393) | (0.0827) |
| *ttloan* | −0.2777 *** | −2.7551 *** | −0.0943 *** |
|  | (0.0412) | (0.1366) | (0.0196) |
| *rota* | 0.0299 | 0.1779 *** | 0.0059 |
|  | (0.0464) | (0.0458) | (0.0448) |
| *depositgrowth* | −0.0034 | −0.0064 *** | −0.0039 |
|  | (0.0026) | (0.0023) | (0.0029) |
| *loan_asset_ratio* | −2.0109 *** | 1.9847 *** | −1.5443 *** |
|  | (0.3685) | (0.4337) | (0.3011) |
| *nplra* | 0.0262 | 0.0336 | 0.0296 |
|  | (0.0395) | (0.0390) | (0.0348) |
| *asset_liability_ratio* | −1.4730 | 6.1451 *** | −0.6516 |
|  | (1.8920) | (1.7693) | (1.7893) |
| Individual fixation | No | Yes | Yes |
| Year fixed | No | Yes | Yes |
| $R^2$ | / | 0.4007 | / |
| Number of samples | 899 | 899 | 899 |

Note: Standard errors are indicated in parentheses, * indicates significant at 10% level, ** indicates significant at 5% level, and *** indicates significant at 1% level. Same as below.

The comparison finds that the *DID* of the implementation effect of direct monetary policy on the credit supply to SMEs is positive with or without the inclusion of individual fixed effects and time fixed effects, as columns (1)–(3) are significant at the 1%, 5%, and

10% significance levels, respectively. All three regressions are significant, implying that the direct monetary policy significantly promotes the supply of bank credit to SMEs, and the direct effect is significant. From the regression results of the control variables in the baseline regression (2), it is found that the *loan_asset_ratio*, *asset_liability_ratio*, and *rota* are all significant at the 1% significance level for the supply of credit to SMEs, and the stronger the asset strength of commercial banks, the more efficient and profitable the capital utilization is. The stronger the asset strength, capital utilization efficiency and profitability of commercial banks, the more effective the implementation of direct monetary policy tools to increase the supply of credit to SMEs.

The *ttloan* and *depositgrowth*, both significant at the 1% significance level, have a negative impact on the supply of credit to SMEs. This indicates that the more deposits commercial banks have, the more the total loans increase and the more funds are available for banks to lend, although SMEs are at a disadvantage due to their own size and limited use of funds, and loans flow more to medium and large enterprises, weakening the share of loans to SMEs. The effect of the bank non-performing loan ratio (*nplra*) on the share of loans to SMEs is positive but not significant, indicating that the level of non-performing loan ratio is not a factor affecting commercial banks' decision to promote credit supply to SMEs. It can also be seen from the Tobit regression results (3) that the regression coefficient of direct monetary policy is significantly positive at the 10% level, which helps to drive the supply of credit to SMEs. The previous empirical results also show that after the implementation of the two direct monetary policy tools, the loan interest rates of SMEs have been reduced, the proportion of loans of SMEs in local corporate banks has increased, and the policy effect is significant [39].

### 6.4. A Test of the Role of Digital Inclusive Finance Regulation

Digital inclusive finance can help commercial banks alleviate the information asymmetry in the credit process, make it easier to identify and meet the credit needs of SMEs, and improve the banks' ability to manage risk for SMEs. At the same time, digital inclusive finance also enables commercial banks to obtain more comprehensive and accurate information on loans to all enterprises, reduce the cost of information collection and matching, simplify the loan approval process, and enhance the applicability of the model, which will ration credit to SMEs compared to large enterprises. Therefore, it is worth exploring whether digital inclusive finance plays a certain moderating role in the process of direct monetary policy influencing banks' credit supply to SMEs.

This paper selects four dimensions, Digital Inclusive Finance Index (*index_aggregate*), digital financial coverage breadth (*coverage_breadth*), digital financial usage depth (*usage_depth*) and digitalization level of financial inclusion (*digitization_level*), to test the moderating role of digital inclusive finance in relation to direct monetary policy in the process of influencing bank credit supply to SMEs. As shown in Table 4, the regression results found that three indicators, Digital Inclusive Finance Index, digital financial coverage and digital financial usage depth, have no significant moderating effect on the credit supply of SMEs and the coefficient is negative, while the coefficient of the digitalization level of inclusive finance is positive but the moderating effect is not significant. The moderating effect of digital inclusive finance on the credit supply of SMEs by direct monetary policy has not been statistically significant yet. The reason may be that the direct monetary policy was introduced late and the amount of accessible data is little, while digital inclusive finance is universal, commercial banks apply technology to the credit process of all enterprises, large enterprises have credit advantages over SMEs, more funds flow to large enterprises, and the credit supply for SMEs is not obvious, i.e., as Hypothesis 2b holds.

**Table 4.** Moderating effect of digital inclusive finance on the supply of credit to SMEs of banks influenced by direct monetary policy.

|  | *index_aggregate* | *coverage_breadth* | *usage_depth* | *digitization_level* |
|---|---|---|---|---|
| *SMEL* | −0.0011 | −0.0004 | −0.0020 | 0.0001 |
|  | (0.0024) | (0.0020) | (0.0021) | (0.0028) |
| $R^2$ | 0.4010 | 0.4008 | 0.4022 | 0.4007 |
| Number of samples | 899 | 899 | 899 | 899 |

## 7. Robustness Test

In order to make the empirical results of the hypotheses as stable as possible, this paper will conduct four robustness tests—namely, replacement econometric model test, adjusted sample period test, placebo test, and adjusted Digital Inclusive Finance Index test.

### 7.1. Replacement Econometric Model Test

A breakpoint regression model (Regression Discontinuity Design, RDD) is used to estimate the policy effects of the implementation of the direct monetary policy. Since the direct monetary policy requires specific local banks to allocate preferential funds to key areas and weak links of the national economy, such as SMEs, the supply of credit to SMEs increases. From Figure 4, it can be found that there is a significant jump in loans to SMEs after 2020, while the supply of bank credit to SMEs increases after the implementation of the direct monetary policy.

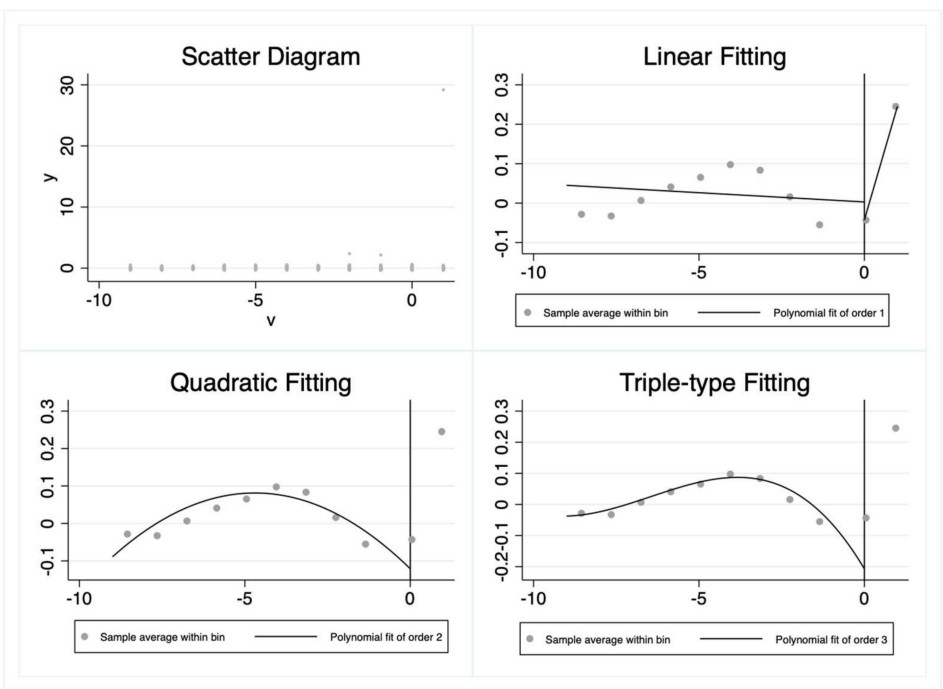

**Figure 4.** Impact of direct monetary policy on the supply of credit to SMEs by banks (RDD).

### 7.2. Adjustment of the Sample Period Test

The period from 2011 to 2021 is the sample period of commercial banks in the previous benchmark regression model. Given the late implementation of the financial inclusion policies and the late establishment of some commercial banks, as well as the large deviation of some indicators from the overall sample mean at the beginning of the establishment, to make the findings more robust, the sample period is replaced with the period from 2015 to 2021 and the *DID* is re-estimated. The regression results are shown in Table 5, where column (1) is the baseline regression without controlling individual and time effects, column (2) is the baseline regression with controlling individual and time effects, and

column (3) is the Tobit regression. The baseline regressions of direct monetary policy are all significant at the 1% level of significance for the increase in the credit supply to SMEs, and the Tobit regression is significant at the 10% level. The results show that research Hypothesis 1 is validated.

**Table 5.** Impact of direct monetary policy on the supply of credit to SMEs by banks (2015–2021).

|  | (1) | (2) | (3) |
|---|---|---|---|
|  | *SMEL* | *SMEL* | *SMEL* |
| *DID* | 0.2886 *** | 0.5367 *** | 0.1700 * |
|  | (0.0840) | (0.1352) | (0.0938) |
| *ttloan* | −0.2957 *** | −4.4377 *** | −0.1047 *** |
|  | (0.0504) | (0.1609) | (0.0247) |
| *rota* | 0.0576 | 0.2042 *** | −0.0150 |
|  | (0.0676) | (0.0529) | (0.0596) |
| *depositgrowth* | −0.0029 | −0.0071 *** | −0.0030 |
|  | (0.0034) | (0.0025) | (0.0038) |
| *loan_asset_ratio* | −2.9338 *** | 3.6094 *** | −2.1791 *** |
|  | (0.5064) | (0.5431) | (0.3957) |
| *nplra* | −0.0076 | 0.0410 | 0.0491 |
|  | (0.0503) | (0.0405) | (0.0422) |
| *asset_liability_ratio* | −5.0008 | −3.6159 | −2.2185 |
|  | (3.1553) | (2.5436) | (2.5551) |
| Individual fixation | No | Yes | Yes |
| Year fixed | No | Yes | Yes |
| $R^2$ | / | 0.6253 | / |
| Number of samples | 711 | 711 | 711 |

Note: Standard errors are indicated in parentheses, * indicates significant at 10% level, and *** indicates significant at 1% level.

### 7.3. Placebo Test

In a counterfactual placebo test, a dummy policy implementation time is constructed to advance the implementation of direct monetary policy to 2019. Using two sample periods of commercial banks, 2011–2021 and 2015–2021, makes the test results more reliable, and the results are shown in columns (1) and (2) in Table 6. Whether using the sample period of 2011–2021 or the sample period of 2015–2021, the implementation of the direct monetary policy instrument does not significantly enhance the credit supply to SMEs, indicating that the credit supply to SMEs by commercial banks does not change significantly with the advancement of the policy implementation time, which can indicate the robustness of the regression results of the benchmark model.

**Table 6.** Impact of direct monetary policy on the supply of credit to SMEs by banks (Placebo Test).

|  | (1) | (2) |
|---|---|---|
|  | *SMEL* | *SMEL* |
| *DID* | −0.0116 | −0.0297 |
|  | (0.0632) | (0.0554) |
| *ttloan* | −2.7110 *** | −4.3702 *** |
|  | (0.1361) | (0.1624) |
| *rota* | 0.1769 *** | 0.2068 *** |
|  | (0.0460) | (0.0537) |
| *depositgrowth* | −0.0066 *** | −0.0075 *** |
|  | (0.0023) | (0.0025) |
| *loan_asset_ratio* | 1.9274 *** | 3.6282 *** |
|  | (0.4359) | (0.5516) |
| *nplra* | 0.0301 | 0.0406 |
|  | (0.0393) | (0.0411) |

**Table 6.** *Cont.*

| | (1) | (2) |
|---|---|---|
| | *SMEL* | *SMEL* |
| *asset_liability_ratio* | 6.5334 *** | −2.2101 |
| | (1.7731) | (2.5615) |
| Individual fixation | Yes | Yes |
| Year fixed | Yes | Yes |
| $R^2$ | 0.3953 | 0.6144 |
| Number of samples | 899 | 711 |

Note: Standard errors are indicated in parentheses, *** indicates significant at 1% level.

## 7.4. Adjustment of the Digital Inclusive Finance Index

The credit index in digital inclusive finance can better reflect the application of digital inclusive finance in the credit process of commercial banks and better measure the credit environment of commercial banks. Therefore, this paper selects the credit index (credit), which is the depth of digital finance usage in the Digital Inclusive Finance Index of Peking University, as an indicator to measure the development level of digital inclusive finance and then conducts robustness tests. The regression results are shown in Table 7, where columns (1)–(4) show the results for the sample period 2011–2021, the sample period 2015–2021, the sample period 2011–2021 for the virtual policy implementation, and the sample period 2015–2021 for the virtual policy implementation, respectively. The relevant results continue to support Hypothesis 2b.

**Table 7.** Moderating effect of digital inclusive finance on direct monetary policy affecting the supply of credit to SMEs by banks.

| | (1) | (2) | (3) | (4) |
|---|---|---|---|---|
| | *SMEL* | *SMEL* | *SMEL* | *SMEL* |
| *credit* | −0.0014 | −0.0001 | −0.0123 ** | −0.0094 * |
| | (0.0039) | (0.0039) | (0.0048) | (0.0050) |
| Individual fixation | Yes | Yes | Yes | Yes |
| Year fixed | Yes | Yes | Yes | Yes |
| $R^2$ | 0.4050 | 0.6260 | 0.4099 | 0.6244 |
| Number of samples | 899 | 711 | 899 | 711 |

Note: Standard errors are indicated in parentheses, * indicates significant at 10% level, ** indicates significant at 5% level.

## 8. Conclusions and Implications

Based on the annual data of domestic listed commercial banks from 2011 to 2021, this paper uses the double-difference model to empirically test the implementation effect of direct monetary policy on the credit supply to SMEs. The adjustment effect model is used to empirically test whether digital inclusive finance plays a regulatory role in the process of direct monetary policy affecting the credit supply to SMEs. The robustness of the empirical results is tested through four aspects: replacement econometric model test, adjusted sample period test, placebo test, and adjusted Digital Inclusive Finance Index test. The comparison finds that the DID of the implementation effect of direct monetary policy on the credit supply to SMEs is positive with or without the inclusion of individual fixed effects and time fixed effects, with all being significant at the 1%, 5%, and 10% significance levels, respectively. All three regressions are significant, implying that the direct monetary policy significantly promotes the supply of bank credit to SMEs, and the direct effect is significant. SMEs affected by the COVID-19 pandemic have found it easier to obtain bank loans through these two direct monetary policies, which has alleviated the financing problem to a certain extent. The three indicators, Digital Inclusive Finance Index, digital financial coverage and digital financial usage depth, have no significant moderating effect on the credit supply to SMEs, and the coefficient is negative, while the coefficient of the digitalization level of

inclusive finance is positive but the moderating effect is not significant. The moderating effect of digital inclusive finance on the credit supply to SMEs by direct monetary policy has not been statistically significant yet. The following policy implications are obtained.

First, it is of great significance to improve the structural monetary policy transmission mechanism. Direct monetary policy tools alleviate the liquidity constraint of commercial banks' credit allocation, although commercial banks still suffer the credit risks and bad debt losses of SMEs. Structural monetary policy tools can add price feedback mechanisms and risk mitigation mechanisms to accurately price the business risks and investment returns of SMEs, and they can compensate for the risks that cannot be priced by the market and the lack of collateral security for SMEs through risk mitigation mechanisms. More targeted use of structural monetary policy tools strengthens incentives for commercial banks' risk-taking and profitability, promotes structural reform of the commercial banking system and credit system, further reduces financing costs for SMEs, and enables monetary policy tools to benefit real enterprises more effectively and directly.

Second, it is of great necessity to establish a sustainable development mechanism for digital inclusive finance and strengthen the use of digital inclusive finance in the credit supply for SMEs. The government researches and formulates policies and plans to support the transformation of digital inclusive finance to support SMEs, strengthens the training of talents in digital inclusive finance, accelerates the research on and innovation in digital inclusive finance technology, establishes credit files for SMEs, allocates financial service resources, and provides a financial information-sharing platform for SMEs. Digital inclusive finance builds an ecological chain for financing SMEs through blockchain and other technologies, establishes a digital monitoring system, realizes refined digital risk management for the entire process of business credit approval, risk warning and post-loan management, and implements more accurate risk assessment for SMEs, so as to empower commercial banks to be "willing to lend, daring to lend, able to lend". SMEs use digital inclusive finance to actively improve their own information technology, enhance their credit availability and reduce financing costs through digital transformation.

Third, it is of great importance to improve the quality of commercial banks' assets and profitability, and to enhance the capital strength to serve SMEs. In addition, it is necessary for policy makers to support commercial banks to supplement capital through multiple channels such as issuing additional shares, placing shares and convertible bonds to enhance the capital strength of banks and improve the quality of bank assets, to improve profitability by improving the disposal method and disposal of non-performing bank assets, to encourage and guide commercial banks to set up additional small and micro branches at the grassroots level, promote the establishment of grassroots inclusive financial organizations, focus on revitalizing the capital stock while continuously increasing the amount of capital, alleviate the phenomenon of commercial banks "shying away from loans", and improve the availability of financial services for SMEs.

Fourth, it is of great importance to continue to promote the construction of the commercial banking system and social credit system. China's commercial banking system, which is dominated by large state-owned banks, and the imperfect social credit system are important reasons for the financing problems of SMEs. We should focus on building a multi-level commercial banking system. While promoting the service focus of large state-owned banks for SMEs, we should support the cultivation of the sustainable development of small and medium-sized banks, build a financing platform for small and micro enterprises and other fields, and organically integrate the capital demand side with financial resources. It is also necessary to speed up the construction of a unified credit system in society, relying on digital inclusive finance to establish credit files and carry out credit portraits for SMEs. Banks and other financial institutions use the above information to innovate credit loan products and implement accurate management in loan approval, post-loan management, risk early warning, etc. A win–win situation for industry financing and a reduced bank credit risk.

The direct monetary policy tool is an important measure for alleviating the problem of difficult and expensive financing for SMEs during the COVID-19 pandemic, which is of great significance to improve monetary policy regulation tools, solve the financing problems of small and medium-sized enterprises in the industrial chain, optimize the allocation of financial resources and improve financial support for the development of the real economy. There are some shortcomings of this paper. This paper only studies the problem of direct monetary policy to alleviate the financing difficulties of SMEs. Due to the difficulty of obtaining the data of loan interest rates for SMEs, the problem of expensive financing for SMEs cannot be studied in depth. At the same time, based on the lack of collateral for SMEs and their own scale limitations, there is a strong correlation between the bank credit supply to SMEs and their credit risks. The regulatory effect of digital inclusive finance on the impact of direct monetary policy on the credit supply of SMEs has not yet been shown. However, in the case of uncertain economic policy, after the introduction of monetary policy, whether or not digital inclusive finance can reduce the risk of small and micro credit so as to increase the credit supply of banks to SMEs can be regarded as one of the future research directions.

**Author Contributions:** R.J.: methodology, data curation, formal analysis, writing—original draft, review, and editing. J.R.: conceptualization, supervision, validation, and writing—review and editing. All authors have read and agreed to the published version of the manuscript.

**Funding:** This work has been supported by the Fundamental Research Funds for the Central Universities, Fund No. 2023YJS109.

**Data Availability Statement:** The raw data supporting the conclusions of this article will be made available by the authors without undue reservation.

**Conflicts of Interest:** The authors declare that the research was conducted in the absence of any commercial or financial relationships that could be construed as potential conflict of interest.

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
