# Peer review of "Does Direct Monetary Policy Affect the Supply of Bank Credit to Small and Medium-Sized Enterprises? An Analysis Based on Chinese Data"

_sustainability, doi:10.3390/su151511674_

Round 1
Reviewer 1 Report
The Introduction fails to motivate the study. In the present form, it resembles a mini-review of literature, rather than discussing any policy-level problem. Why this study is necessary? What policy level problem this study is addressing? How is the study expected to provide any solution to that problem? How does the choice of sample is complementing that problem? Are the results and policies generalizable? The introduction is silent in all these aspects. The mere choice of new variables, or choosing a new context is not considered as a contribution of a study.
What is the aim of the review of literature? The authors have merely listed out the studies without even creating a debate among them. Without that debate and thoughtful contradictions, the research gap cannot be substantiated. Moreover, the research gap is not clear. It should be policy-driven, not method/variable/context-driven.
No particular economic theory is discussed to link the studied variables. The authors require to clarify the choice of methods applied in the study. The policy implications are general not based on the findings. The limitation and future research directions are also missing.
The language quality of the paper is below for publication
Reviewer 2 Report
Introduction – the authors have to clarify the purpose of the paper and better state the need for such a research.
Literature Review – the authors explain the impact of structural monetary policy tools and digital inclusive finance on the supply of credit to SMEs, and the way in which digital inclusive finance affects the transmission of direct monetary policy. But this part must include similar or comparative research papers explaining their methods and results. The presented research need is not sufficiently linked to the previous text in order to specify the knowledge gap.
The hypotheses are not sufficiently built on theoretical analysis, but derive from the direct monetary policy mechanism of The People's Bank Of China. They must be developed on the basis of existing theoretical developments.
The conclusions of the empirical analysis are too little highlighted. For the most part, the conclusion of the paper includes policy recommendations, which do not follow from the analysis undertaken in the paper.
Reviewer 3 Report
I appreciate the authors' efforts in presenting the article titled "Does Direct Monetary Policy Affect the Supply of Bank Credit to SMEs? An Analysis Based on Chinese Data." While I found the article to be insightful, I have a few considerations and constructive comments to offer. I hope these suggestions will assist the authors in enhancing their research further.
Abstract:
The abstract provides a clear and concise summary of the research objectives, methodology, findings, and suggestions for further improvement. It effectively communicates the main points of the study and highlights the importance of supporting SMEs for the development of the real economy.
Section 1:
The introduction provides a comprehensive overview of the importance of SMEs in China's economy and the role of banks in supporting their development, emphasizing the need for increased support and the relevance of digital inclusive finance.
However, the introduction could benefit from improved organization and structure, clearer connections between objectives and subsequent sections, and addressing the limitations and potential biases of the study's data collection and analysis to enhance its overall quality.
I am satisfied with the content and organization of Sections 2, 3, 4, 5, 6, and 7. The details provided in these sections are comprehensive and effectively support the study's objectives. The structure is logical and allows for a clear progression of ideas. Well done!
Section 8:
The conclusion and implications section provides a clear summary of the study's findings and offers valuable policy implications. The authors effectively highlight the significance of improving the structural monetary policy transmission mechanism, establishing a sustainable development mechanism for digital inclusive finance, and enhancing the quality of commercial banks' assets and profitability to better support SMEs. However, the section can be further enhanced by addressing certain areas.
Suggestions for Improvement:
Provide more specific details: The authors could provide more specific details regarding the proposed improvements to the structural monetary policy transmission mechanism and the establishment of a sustainable development mechanism for digital inclusive finance. This would help readers better understand the practical steps and strategies to be implemented.
Consider addressing limitations and potential biases: It would be beneficial for the authors to acknowledge and address any limitations or potential biases in their study. This could include discussing any data limitations, methodological constraints, or potential external factors that may have influenced the findings. Addressing these aspects would strengthen the overall validity and reliability of the study.
Include future research directions: It could be valuable for the authors to suggest potential areas for future research based on their findings. This would provide readers with a sense of continuity and inspire further exploration of the topic.
Clarify the relationship between direct monetary policy and COVID-19: The authors briefly mention the importance of direct monetary policy in alleviating financing problems for SMEs during the COVID-19 pandemic. However, it would be helpful to clarify the specific linkages and implications of direct monetary policy in the context of the pandemic, especially in relation to the study's findings.
Highlight practical implications: While the policy implications are mentioned, the authors could further emphasize the practical implications for policymakers, financial institutions, and SMEs. Exploring specific actions and strategies that can be implemented based on the findings would enhance the section's practical relevance.
Reviewer 4 Report
The research exploring how the Direct Monetary Policy Affect the Supply of Bank Credit to small and medium-size enterprises seems relevant and contemporary; however, I have specific concerns with the manuscript in its current form:
My major concerns are listed below:
1. The authors need to present in a more clear way the contribution of this research, given the large body of existing literature on this topic. My question: what is novel?
2. In the introduction it is not clear ‘what was done’ to address the research objective? What is the theory? What are the methodology and methods implicated in this paper? These need to be clearly mentioned in the introduction.
3. Regarding the Numerical Study, I don’t have much to add. The empirical analysis although interesting, but I would strongly suggest, that the author/s add more intuitive explanation linking;
4. The art of crafting a good story is very important so you need to work on this to make it easy for the reader to understand and to make it engaging to read. I hope my comments are helpful in this regard;
5. I suggest that the authors restructure the paper and establish the novelty of the research.
I suggest a careful reading of the paper.
Reviewer 5 Report
This paper investigates how direct monetary policy affects the supply of bank credit to SMEs. The paper is fairly well written. The introduction to the paper is good and the authors have also conducted a comprehensive literature review.
However, Sections 4 and 5 need some improvement. In Section 4, since more than one hypothesis is tested, the title of Section 4 should be changed to reflect that. In Section 5, authors have stated that the data were collected manually from the annual reports of 174 Chinese listed commercial banks from 2011 to 2021. However, it is not very clear how many banks were included in the study. According to Table 2 (Line 441), authors have used 910 observations in estimating the models. If 11 years of data were used, authors may have used close to 83 banks. In addition, though authors have not specifically stated, models appear to be estimated using panel data methodology. If the panel data methodology has been used, Models 1 and 2 (in Line 385 and Line 388) should be changed to include two subscripts, one for time and another for the cross-sections. In addition, two variables included in Model 2, namely, df and DID*df, do not have slope coefficients.
In Table 1, variable descriptions are missing for some variables. A note should be included in Tables 4-7, specifying what figures in parentheses represent. In addition, authors have also not compared the findings of this study with that of previous studies.
Nevertheless, the conclusions adequately tie together the other elements of the paper. To some extent, the paper has expressed its case, measured against the technical language of the field and the expected knowledge of the journal's readership.
The quality of English language is good.
Reviewer 6 Report
The paper tackles a globally important issue, namely whether direct monetary policy can stimulate real economic dynamism via fostering SMEs’ credit-based development. The paper’s merit is at least two-fold: it goes beyond the traditional SME policies by concentrating on direct monetary policy (e.g., digital inclusive finance); second, in doing so, it addresses an important case, the Chinese economy where direct monetary policy measures were introduced, whereby it is based on a new sample.
The paper is of high relevance in the developed world as well where the excessive financialisation at the expense of the real economy has been being with us for decades, when large concentrations are dominating and the financial sector / including the banking system seem to have forgotten the middle-sphere (SMEs as well).
Authors shall also address some potential unintended consequences (at least by mentioning a few of them):
· Rising credit-flow should be accompanied with (more spectacular) productivity improvements, if not, the case of cheap money will create a new culture for businesses in which they are not competing with each other in a dedicated way but surviving even bad performances (e.g., zombie phenomena);
· There must be an exit strategy for such policy intervention otherwise, in the medium and longer run, SMEs start to become laxer when it comes to prudent and innovative approaches
· China has implemented so far monetary easing which proved to be less effective as households and private firms built up savings and reduced borrowing and spending to repair balance sheets after three years of COVID curbs. Monetary policy and even fiscal policy shall be In tandem to curb inflationary pressure (stimulus to increase SMEs investments that are probably leading to higher prices while people started to return to spending after COVID years which will also prop up prices) while trying to unleash private savings to spark real economic dynamism thorough entrepreneurship;
Round 2
Reviewer 1 Report
The authors adressed all the comments.
Minor editing of English language required.
Reviewer 2 Report
Introduction – the authors has clarified the purpose of the paper and the need for such a research.
Literature Review – the authors explained the impact of structural monetary policy tools and digital inclusive finance on the supply of credit to SMEs, and the way in which digital inclusive finance affects the transmission of direct monetary policy. The authors included comparative research papers explaining their methods and results. The presented research is linked to the previous text.
The hypotheses are built on theoretical analysis.
The conclusions of the empirical analysis are better highlighted.
Reviewer 4 Report
Congratulations to the authors for addressing successfully all my comments.
Reviewer 5 Report
Authors have addressed all my concerns about the previous version of the manuscript.
Quality of English language is good.